

# Structural changes of CAST soot during a thermal-optical measurement protocol

Theresa Haller[1], Christian Rentenberger[1], Jannik C. Meyer[1], Laura Felgitsch[2], Hinrich Grothe[2], Regina Hitzenberger[1]

[1]University of Vienna, Faculty of Physics, Vienna, 1090, Austria

[2]Vienna University of Technology, Institute of Materials Chemistry, Vienna, 1060, Austria

Correspondence to: Regina.Hitzenberger@univie.ac.at

## Abstract

Thermal-optical measurement techniques are widely used to classify carbonaceous material. The results of different methods for total carbon are comparable, but can vary by >44% for elemental carbon. One major cause of variation is the formation of pyrolyzed carbon during the heating process which occurs mainly in samples with a high amount of brown carbon (BrC). In this study the structural changes of two different CAST aerosol samples caused by the heating procedure in a thermal-optical instrument were investigated with UV-VIS and Raman spectroscopy, the Integrating Sphere technique and transmission electron microscopy. All analysis techniques showed significant structural changes for BrC rich samples at the highest temperature level (870°C) in helium. The structure of the heated BrC-rich sample resembles the structure of an unheated BrC-poor sample. Heating the BrC rich sample to 870°C increases the graphitic domain size within the material from 1.6 nm to 2 nm. Although the Raman spectra unambiguously show this increase of ordering only at the highest temperature step, UV-VIS and IS analyses show a continuous change of the optical properties also at lower temperatures. The sample with a negligible amount of BrC, however, did not show any significant structural changes during the whole heating procedure.



# 1 Introduction

Carbonaceous material contributes a large amount to atmospheric aerosols from 20% of alpine PM2.5 up to 40% of urban PM2.5 (Pöschl, 2005) and up to 50% in PM10 (Yttri et al., 2007). Major sources include incomplete combustion of fossil and biogenic fuels, emission of primary particles by the biosphere and gas-to-particle conversion of precursor gases emitted by the biosphere or anthropogenic activities (IPCC 2013; Després et al., 2012; Zhu et al., 2018; McNeill, 2017). Carbon containing aerosols influence the global radiation balance due to their optical properties and their capability to act as cloud condensation nuclei and ice nucleating particles. Particularly black carbon (BC) has an important effect on radiative forcing as it has the strongest light absorption of all aerosol components (Bond and Bergstrom, 2006; IPCC 2013). Carbonaceous material has also been found to cause various deleterious health effects (Highwood et al., 2006; Anderson et al., 2012; Mesquita et al., 2017; EEA 2017; Partanen et al., 2018).

Depending on its origin carbonaceous material ranges from agglomerations of primary spherical particles with a graphitic like internal structure to non-ordered organic material. While primary carbonaceous particles produced under well-defined combustion conditions are rather homogeneous, real-atmosphere carbonaceous particles are internal mixtures of different carbonaceous and non-carbonaceous materials of various origins (e.g. Okada and Hitzenberger, 2001; Zhang et al., 2014; Ye et al., 2018; IPCC, 2013; Deboudt et al., 2010; Pratt and Prather, 2010; Bondy et al., 2018). As carbonaceous aerosol material plays such an important role and is so diverse, the correct determination of carbonaceous fractions in the aerosol is essential. However, this endeavour is not trivial because measurement techniques are influenced not only by the properties of the material itself, but also by the non-carbonaceous materials associated with the particles and their mixing state.

During the past decades, numerous methods were developed and intercomparison studies of different methods applied to both laboratory-generated and ambient aerosol were performed, which showed similarities and discrepancies of the results of different methods depending on sources and aerosol constituents (see e.g. the overviews given by Watson et al., 2005; Venkatachari et al., 2006; Müller et al., 2004).

Analysis techniques can be grouped broadly into optical techniques, which utilize light absorption and its wavelength dependence and thermal techniques which separate the material based on its thermal properties. Usually, the term black carbon (BC) is used for the carbonaceous fraction with high light absorption in the whole visible wavelength range, and brown carbon (BrC) for the carbonaceous fraction with a pronounced wavelength dependent absorption with high absorption in the blue and weak absorption in the red part of the spectrum. Thermal techniques separate carbonaceous material into organic carbon (OC, thermally unstable) and elemental carbon (EC, thermally refractory).The latter oxidizes in air at temperatures above 600°C (Andreae and Gelencsér, 2006) and does not evaporate in the absence of oxygen below fairly high temperatures (the definitions range from 550°C (Bond and Bergstrom, 2006) to 700°C (Chow et al., 2004)). EC and BC are often loosely used as synonymous, as both are strongly light absorbing, but the comparability of values measured with different techniques depends on the composition of the aerosol and the measurement method (Cheng et al., 2012; Cavalli et al., 2010).Optically determined BrC is a subfraction of thermally determined OC, and originates mainly from combustion of





biomass or biofuels (Mayol-Bracero et al., 2001; Hoffer et al., 2006; Schmidl et al., 2008; Park et al., 2018; Fan et al., 2016; Park and Yu, 2016; Sun et al., 2017).The thermal and optical behaviour of carbonaceous material is caused by its internal structure (i.e. chemical composition, ordering and bonding types). As outlined by Pöschl (2005) there is no sharp boundary between the thermally unstable OC and EC respectively BC but a continuous transition from organic carbon to elemental

carbon in terms of structure as well as of thermal and optical properties: The higher the degree of graphitization, the higher is the thermal stability and the broader the wavelength range of light absorption.

In order to account for pyrolyzation of OC during the heating process used in the thermal techniques, thermal-optical techniques were developed.  In these techniques the sample is heated stepwise first in an inert He atmosphere to a maximum temperature, which depends on the measurement protocol (see below), then oxygen is added to the carrier gas and the

sample is further heated. The amount of carbon leaving the filter at each temperature step is detected. The darkening of the sample due to pyrolysis is traced by monitoring a transmission/reflection laser signal. After the addition of oxygen both the originally present EC and the pyrolyzed carbon (PC) are oxidized during the successive heating steps. All carbon leaving the filter until the laser signal has reached its original value is assumed to originate from PC and is attributed to the OC fraction in the subsequent evaluation.

The crucial assumption for this laser correction is that either PC has the same molecular structure as the original EC or that PC burns off completely before EC is oxidized.  None of these conditions is fulfilled, as was shown by Yu and Yang (2002), who found that in many cases the light absorption coefficient (σ) of the two fractions is not the same at least at the wavelength used in their thermal-optical instrument (680 nm). Moreover, the value of σ of PC is not constant even during a single thermal analysis. This uncertainty in accounting for PC leads to uncertainties in the OC/EC split, which is the subject

of numerous instrument intercomparison studies (see e.g. the overview given by Cavalli et al., 2010).

Different thermal-optical measurement protocols vary in height and duration of the individual temperature steps, particularly in the maximum temperature of the inert (He) mode (maximum temperatures between 580°C (IMPROVE A, Chow et al., 2004) and 870°C (NIOSH870, Panteliadis et al., 2015; Birch and Cary, 1996)). This leads to different charring behaviours of OC. As a consequence the OC/EC split varies with the temperature protocol, while the total carbon (TC, i. e. the sum of OC,

EC and, if present, carbonate carbon) values usually are quite comparable. Different thermal methods agree within 5-15% in the amount of TC but vary up to 44% for EC (Reisinger et al., 2008; Cavalli et al., 2010; Yu and Yang, 2002; Cheng et al., 2012; Hitzenberger et al., 2006; Panteliadis et al., 2015).

The largest discrepancies between thermal-optical methods applied to atmospheric samples occur in the presence of appreciable amounts of BrC (e.g. Wonaschuetz et al. 2009; Reisinger et al., 2008). Kim et al. (2014) compared OC and EC

values obtained with two different thermal-optical protocols (NIOSH and EUSAAR) for laboratory generated soot from propane combustion (produced by a miniCASTburner.  For EC/TC ratios > 0.5, the differences of EC/TC between both methods were below 15%. For lower EC/TC ratios when significant amounts of BrC (measured with an optical method) were present, larger differences (30%) are noticeable.



Besides the presence of BrC, metal salts were also found to affect the OC/EC split in thermal-optical methods by enhancing the charring of OC, and/or by reducing the oxidation temperature of EC (Wang et al., 2010; Bladt et al., 2014).

The aim of the present study was to shed more light on the processes which lead to the discrepancies between measurement methods using different thermal-optical protocols. The formation of PC during thermal-optical measurements is not yet fully

understood. The purpose was to investigate the structural changes of carbonaceous aerosol samples occurring during pyrolysis. In order to exclude the effects of other, non-carbonaceous material on the charring process, the structure of two different types of laboratory generated soot – one with a high tendency to form PC and one with a negligible tendency to form PC – was analysed in different stages of pyrolyzation in order to obtain more information about the internal structure of PC and its differences or analogies to the structure of EC. A thermal-optical EC-OC analyzer (Sunset Instruments Inc.,

description see below) was used not only for the analysis of these soot samples but also as a means for sample preparation in order to obtain pyrolyzed samples from each heating step that had been exposed to the same atmospheres, heating rates and temperature plateaus as during the thermal-optical analysis. This approach permits an investigation of the structural changes of the material as they occur during the thermal-optical heating procedure.

## 2 Properties of soot

### 2.1 Structure of soot

The term soot is usually used for the product of incomplete combustion or pyrolysis of fossil fuels or organic materials (Andreae and Gelencsér, 2006; Pöschl et al., 2005; Bond and Bergstrom, 2006).

The structure of soot depends on the fuel and on the combustion conditions. Soot produced under oxygen rich combustion conditions and high temperatures consists of agglomerated primary particles with sizes between 10-30 nm (Sadezky et al.,

2005). The primary particles consist of onion like turbostratically ordered graphitic layers. The graphitic domains typically include 3-4 graphene layers and have an extent of about 3 nm with interlayer distances of about 3.5 Å (Sadezky et al., 2005) which is larger than the interlayer distance of an ideal graphitic lattice (3.35 Å) because of the turbostratic arrangement. The atoms in graphitic like material are bound with π-bonds, which form conjugated systems and hence long range orbitals. As the optical gap is small (≈0.5 eV for disordered graphitic layers; Robertson and O'Reilly, 1986), light from a broad spectral

range is absorbed (Bond and Bergstrom, 2006; Chhowalla et al., 1997).

In contrast to this highly ordered type of soot, the molecular structure of BrC is similar to that of polycyclic hydrocarbons (PAH) or humic like substances (HULIS) (Graber and Rudich, 2006; Sun et al., 2007). Due to the smaller expansion of the molecular orbitals there is a larger optical gap between the filled valence band and the unfilled conduction band, which leads to a decreasing absorption efficiency towards the long-wave part of the visible spectrum (Bond, 2001; Kim et al., 2015;

Andreae and Gelencsér, 2006).



## 2.2 CAST soot

For the present study, a miniCAST burner was used to produce differently structured soot by varying the combustion conditions. A detailed description of the miniCAST burner is given below. Different fuel to air ratios in the flame lead to different compositions and structures of the produced particles. According to Kim et al. (2015) and Mamakos et al. (2013)

the particles obtained under oxygen rich settings are comparable to diesel exhaust aerosols. They show a high EC fraction and form relatively large agglomerates with mobility diameters between 70 and 130 nm, which are composed of small (25-30 nm) spherical primary particles (Kim et al., 2015). Under oxygen-poor combustion conditions, spherical particles with sizes in the range 10-60 nm and a low EC fraction (Kim et al., 2015) are formed. While soot from biomass burning contains also ionic components such as $Na^+$, $NH_4^+$, $Ca^{2+}$, $Mg^{2+}$, $K^+$, $Cl^-$, $NO_3^-$ and $SO_4^{2-}$ which could lead to catalysis effects in the

thermal-optical analysis (Ichikawa and Naito, 2017; Wang et al., 2010), these compounds are negligible in soot produced by the CAST burner since only pure propane and air contribute to the combustion.

## 2.3 Basics of Raman spectroscopy of soot

Raman scattering has been used for the structural investigation of soot for several years (Rosen and Novakov, 1987; Sadezky

et al., 2005; Ferrari and Robertson, 2000; Schmid et al., 2011; Knauer et al., 2009; Ivleva et al., 2007a, 2007b). It is sensitive to different C-C bonding types (e.g. graphitic structures) in a material (Sadezky et al., 2005) and depends on the ordering of $sp^2$ sites (Ferrari and Robertson, 2000).

Ideal graphite shows a single peak at a Raman shift of ≈1580 cm$^{-1}$ (G-Peak or graphitic peak), which is related to the ideal hexagonal environment in the extended graphene layers ($E_{2g}$ symmetry). For non-ideal graphite, an additional peak at ≈1350

20   cm$^{-1}$ (D peak or defect peak) appears, which is due to impurities and/or smaller graphene layers and the thus increased number of layer edges. Due to the missing neighbour atoms at these edges the symmetry of the C-C vibration is reduced to $A_{1g}$. When the ordering in the material decreases – which is the case in soot – the two formerly sharp peaks broaden and overlap (see also Fig. 5 and 6) as a consequence of more than two overlapping Raman bands. To separate the contributions of the bands, different authors suggest different curve fitting methods with up to five signals with either Gaussian or

Lorentzian shape (Dippel et al., 1999; Jawhari et al., 1995; Sze et al., 2001; Cuesta et al., 1994). A comparison study performed by Sadezky et al. (2005) found that a five curve fit with four Lorentzian and one Gaussian curves represents most of their spectra best. The curve shapes, band positions and related vibration modes are listed in Table 1.

The ratio of the intensities of the D and G peak ($I_D/I_G$) can be linked to the crystallite size ($L_a$) within the investigated material (Ferrari and Robertson, 2000; Tuinistra and Koenig, 1970). Ferrari and Robertson (2000) propose an increase of $L_a$

with increasing $I_D/I_G$ below $L_a≈2nm$. In this regime $I_D/I_G$ is proportional to the probability to find six-fold aromatic rings in the cluster (Ferrari and Robertson, 2000). For larger crystallite sizes $I_D/I_G$ decreases with increase of $L_a$ (Tunistra and Koenig, 1970). $I_D/I_G$ shows a maximum at $L_a≈2nm$ but has a broad transition regime between the increasing and decreasing



branch. Zickler et al. (2006) compare the $I_D/I_G$ ratio of Raman spectra of charcoal obtained from spruce wood with the crystallite sizes obtained from X-ray diffraction and confirm the proposal of Ferrari and Robertson experimentally. Based on these findings, several authors (Commodo et al., 2016; Ess et al., 2016; Ivleva et al., 2007) use increasing $I_D/I_G$ ratios as criteria to detect an increasing degree of ordering in soot.

## 3 Experimental Setup

For the production of well-defined combustion soot, a **C**ombustion **A**erosol **ST**andard soot generator (type miniCAST 5201C Jing-CAST Technologies, www.sootgenerator.com) was used. In this burner, soot is produced in a propane and air co-flow diffusion flame which was quenched with nitrogen directly after the combustion zone in order to prevent further
combustion processes. The particle flow is subsequently diluted with air. The burning conditions were varied by setting different air to fuel ratios to produce differently structured soot.

Prior to this study, samples were obtained for a variety of air-to-fuel ratios and analysed for BC and BrC with the IS technique (see below). Based on these experiments, two different combustion conditions were chosen here to produce two types of soot: one representing a BrC-rich sample ("brown") and the other a BrC-poor sample ("black") (Table 2). The
samples have a completely different morphology, as is shown below in Fig. 9 and 10.

For sampling, quartz fiber filters (Pall Tissuquartz 2500 QAT-UP, 47mm) were used. The filters had been prebaked at 450°C for an hour and stored for at least 24 hours in a water vapour saturated atmosphere before sampling to prevent adsorption of volatile organic substances during handling (Jankowski et al., 2008). The loaded filters were stored at -22°C except during analysis and further sample preparation (see below). The setup of the sampling system is shown in Figure 1.
Part of the exhaust stream of the miniCAST was diverted and drawn through two parallel filters placed in identical holders. The total flow in this sampling line was controlled by a critical orifice (12.5 l/min). The aerosol flow was diluted with 5 l/min air filtered with a HEPA filter, except for samples to be analysed with UV-VIS spectroscopy, where the flow had to be diluted with 6 l/min air to prevent too dark filter deposits. The dilution air was regulated with a needle valve and measured with a rotameter. For each of the following measurement techniques (UV-VIS, Raman spectroscopy, IS, TEM) a set of eight
1.5 cm$^2$ filter punches was taken from the filters. For the preparation of the heated samples, the EC-OC analyzer (description see below) was programmed according to the NIOSH870 protocol and used as an oven. Each filter punch was inserted into the analyzer and underwent part of the NIOSH870 heating procedure, from the beginning to one of the first eight prescribed temperature levels (see Table 2). At that point the automated heating procedure was interrupted and the oven cooled down to below 75°C in Helium. Then the punches where removed from the instrument and transferred directly into Petri dishes. The
dishes were closed with Parafilm strips and stored at -22°C until the further measurements. This procedure was performed for four sets of filter punches to be analysed with the different techniques.



## 4 Analysis Techniques

### 4.1 Thermal-optical measurements

For the analysis of the filter samples and the heating of the samples a dual-optics EC-OC analyzer (Sunset Instruments Inc.) was used. In the first heating steps, the samples are heated in helium. In the second part of the analysis the samples are exposed to an oxidizing atmosphere consisting of 2% oxygen and 98% helium. Reflectance and transmittance signals of laser light (635 nm) are used to monitor the darkening of the sample caused by pyrolysis during heating in the inert atmosphere. In this study, the NIOSH870 protocol was used. Temperature steps and residence times are listed in Table 3.

The NIOSH870 protocol was chosen here to investigate the structural changes of soot during pyrolysis, because of the strong charring occurring in the He mode at the high temperatures. The EUSAAR2 protocol (Cavalli et al., 2010) which is now used widely in the EU, minimizes charring and is therefore less suitable for the investigation of PC.

### 4.2 Integrating Sphere (IS)

BC and BrC (often taken together as **L**ight **A**bsorbing **C**arbon, LAC) of the original and the heated samples were analysed with an extension of the original integrating sphere technique (described e.g. by Hitzenberger and Tohno, 2001). In this technique, a sample (e.g. a filter punch) is immersed in a liquid and introduced into the center of an integrating sphere illuminated with diffuse light. In our study, a 6 inch integrating sphere (Labsphere, Inc.) coated internally with a nearly ideally diffusely reflective (>99%) material (Spectraflect™) was used. Samples were immersed in a mixture of 10% isopropanol, 40% $H_2O$ and 50% acetone in PE vials. In the extended technique (Wonaschuetz et al., 2009) the sphere is illuminated with a halogen light source equipped with three interference filters (450, 550 and 650 nm) and the wavelength dependent light signal is recorded with a photodiode. The contributions of BC and BrC to the absorption signal are separated in an iterative procedure using calibration curves obtained with a proxy for BC (Elftex 124, Cabot Corp.) and a proxy for BrC (humic acid sodium salt, Acros Organics, no. 68131044).

### 4.3 Raman Spectroscopy

In this study a confocal Raman microscope (Horiba Jobin-Yvon LabRAM 800HR) was used. The Raman microscope was equipped with a 632.8 nm HeNe-laser (maximum output <20 mW) and a CCD camera (Peltier cooled at -60°C). The laser beam was focused on the sample with a 20x magnification objective (CF Plan, 20x/0.35, WD 20.5mm, Nikon). The spectra were calibrated with the Rayleigh line at 0 cm$^{-1}$ and the silicon peak at 521 cm$^{-1}$. The laser power was reduced to 10% for analysing black samples and 25% for brown samples to prevent thermal destruction of the material. All instrument settings (grating 300 lines/mm (3-4 cm$^{-1}$ resolution), confocal hole 1000 μm, acquisition time 5 sec with 30 accumulations) were chosen after a set of test measurements to provide the best signal to noise ratio for the analysis. As the samples burned off partially at the 775°C temperature step in the EC-OC analyzer, the Raman spectra were recorded only for the first seven temperature steps.





The spectra were recorded in the range from 200 cm$^{-1}$ to 2000 cm$^{-1}$ at four measurement points at three positions for each sample to account for possible variations within a filter sample. The averaged spectra for each filter were analysed using OriginPro2016 which has built in functions for baseline correction. Baselines were fitted with a B-spline function based on approximately 20 anchor points. After subtraction of the baseline, the spectra were fitted with five curves (four Lorentzian,

one Gaussian; see Table 1). As initial values for the fit, the band positions obtained by Sadezky et al. (2005) and listed above (Table 1) were used. The standard deviations of the mean spectra were used in the fitting software as weighting of the fit.

### 4.4 TEM – Transmission electron microscopy

A 200kV transmission electron microscope (TEM) Philips CM200 equipped with a Gatan$^{TM}$ Orius CCD camera was used to analyze the nanostructure of the original and the heated samples. Single filter fibres were separated from the samples and

placed onto holey carbon films or between copper grids to facilitate studying freestanding soot particles. High resolution microscopic images and diffraction patterns of particles deposited at the lateral edges of the fibres were taken. Intensity profiles of the diffraction patterns were obtained by both azimuthal integration along rings and background correction using PASAD-tools (Gammer et al., 2010). For comparison a simulated profile of graphite was calculated with JEMS software (Stadelmann, 2004).

### 4.5 UV-VIS Spectroscopy

A LAMDA 750 UV-VIS spectrometer (Perkin Elmer, Waltham, Massachusetts, US), equipped with a tungsten and a deuterium lamp was used to measure the diffuse reflectance of the samples. Reflectance measurements were carried out from 800 to 200 nm with an interval of 1 nm. At 319.2 nm the instrument switches from the tungsten (visible) to the deuterium (UV) lamp. For the measurement the original and heated samples were put into quartz vials without any liquid. The diffusely

reflected light from the filter punches was collected with a 60 mm integrating sphere and detected with a photomultiplier. Absorbance spectra where calculated with the Kubelka-Munk equation (Kubelka and Munk, 1931) from the measured reflectance spectra. Although the Kubelka-Munk theory was developed originally for powder samples, its applicability to aerosol filter samples was shown by Aryal et al. (2014).

Absorbance spectra were calculated from the measured reflection signals $R_\infty(\lambda)$ using the Kubelka-Munk function

$$\text{K-M}(\lambda) = \frac{(1 - R_\infty(\lambda))^2}{2R_\infty(\lambda)} \qquad \text{(Eq. 1)}$$

where K-M($\lambda$) is proportional to the absorbance, assuming infinite sample thickness. The spectra were corrected for background signals obtained from three filter blanks.



## 5 Results and Discussion

### 5.1 Thermal-optical analysis

In the thermal-optical analysis most of the carbon of the "black" sample evolves in the He+$O_2$ phase. Throughout the successive heating steps, the laser reflectance signal remains relatively constant until EC oxidizes (Fig 2). On the other hand,

most of the carbon of the "brown" sample evolves in the He-phase (Fig 3). The laser reflectance signal decreases during the first three temperature steps, indicating a pyrolysis of the organic material. The signal starts to increase slightly at the last temperature step (870°C) in He and increases rapidly at 625°C after $O_2$ is added indicating the combustion of initial EC and/or pyrolyzed OC (PC). The laser signal reaches its initial value at the end of the 700°C temperature step, therefore approximately half of the carbon signal in the oxidizing atmosphere is assigned to OC following the normal procedure of the

thermal-optical analysis method. Results from EC and OC measurements are summarised in Table 3. Figures 2 and 3 show full thermograms of a "black" and a "brown" sample.

### 5.2 Black and Brown carbon

Figure 4 shows the change of BC and BrC during the thermal-optical analysis. The changing BC content of the sample is

qualitatively consistent with the interpretation of the laser signal in the thermal-optical analysis: BC in the "black" sample decreases somewhat during the whole analysis cycle, while BrC is below the detection limit for each temperature step.

For the "brown" sample, BrC decreases continuously during the whole He mode, while BC increases over the He mode up till 870°C (He) and decreases in the He+$O_2$ mode. These findings confirm that BC is built during the thermal-optical analysis of an organic carbon containing sample.

Table 4 summarizes the composition of the original "brown" and "black" samples regarding their EC, OC, BC and BrC content: The "black" sample consists mainly of EC or BC with a negligible amount of BrC. On the contrary, the "brown" sample contains only about 10% EC or BC and approximately 90% BrC.

### 5.3 Raman Spectra

The change of the Raman spectra due to the heating process in the thermal-optical instrument is shown in Figures 5 and 6.

For a better comparison the spectra are normalized to the maximum of the G peak. While the maxima of the D and G peak do not change for the "black" sample, the spectra of the "brown" sample show a relative increase of the D peak for samples heated at 870°C in He. The peak does not increase continuously over the whole heating process, which indicates a significant structural change especially at this temperature step. Following the interpretation of Ferrari and Robertson (2000) and its application to soot samples by Commodo et al. (2016), Ess et al. (2016) and Ivleva et al. (2007), this increase of the D peak

indicates an increased amount of polyaromatic rings in the sample which can be associated with a higher degree of ordering.



For a better comparison, the spectra of the "black" and "brown" original samples and samples heated to 870°C in He are shown in Figure 7. The D peak of the "brown" heated sample reaches nearly the height of the D peak of the "black" original and heated samples.

Figure 8 shows the five-curve-fits of these four spectra. All fits were performed on the smoothed and averaged versions of the spectra. It is obvious that there is no significant change in the five peaks between the original and heated "black" sample. However, the spectra of the "brown" sample show a significant change at the $D4_{fit}$ peak ($\approx 1200$ cm$^{-1}$) and the $D1_{fit}$ peak ($\approx 1350$ cm$^{-1}$), when the sample is heated to 870°C in He: while the $D4_{fit}$ peak (which is related to C=C double bonds, Sadezky et al., 2005) decreases, the $D1_{fit}$ peak (related to graphene layer edges, Sadezky et al., 2005) increases. This implies, that the material in the heated sample contains fewer C=C double bonds and more graphene layer edges compared to the original "brown" sample.

Possible explanations could be a decomposition and evolution of molecules with C=C bonds (e.g. polyenes) and a coincident fragmentation of pre-existing graphene layers or a transformation of molecules with C=C bonds into new small graphene layers. Both scenarios would lead to a higher amount of graphene layer edges (and an increased $D1_{fit}$ peak) and a lower amount of C=C bonds (and a decreased $D4_{fit}$ peak).

TEM images (see below) show, that the ordering in the "brown" sample increases, when it is heated to 870°C in He. The UV-VIS spectra (see below) show a decreasing absorption coefficient which requires a growth instead of a fragmentation of the conjugated orbitals in the material. Both findings are more consistent with the second process.

**5.4 TEM – Transmission electron microscopy**

The TEM images of the "black" and "brown" samples show entirely different morphologies. The soot in the "black" sample consists of agglomerates of spherical primary particles with diameters of about 20 nm (Fig. 9). The primary particles show an onion like graphitic structure as it is also reported by Sadezky et al. (2005) and Kim et al. (2015). In contrast, the "brown" sample does not show any ordered internal structure but consists of an oily substance (described as "condensed organic species" by Moore et al., 2014) which adheres well to the filter fibres (Fig. 10).

After heating to 870°C under helium atmosphere, the "black" sample shows the same graphitic like structure as the original "black" sample. However, the "brown" heated (870°C He) sample seems to have a more ordered internal structure in the form of layers than its original version as indicated by stronger noticeable local fringe-like contrast (Fig. 11).

The electron diffraction patterns (Fig. 12) of both heated and original samples show rings with maxima at 2.8 nm$^{-1}$ (A), 4.9 nm$^{-1}$ (B) and 8.4 nm$^{-1}$ (C). The positions of the rings are very similar to those of simulated graphite with randomly oriented small graphitic domains (cf. inset in Fig. 13). Therefore we conclude that the ring A at 2.8 nm$^{-1}$ can be related to the layer distance of graphite and the ring B at 4.9 nm$^{-1}$ to the (100) or (101) planes of graphite. All rings appear in the diffraction patterns of both the "black" and the "brown" samples in Fig. 12 but they are broader for the "brown" original sample. The latter result is demonstrated more clearly in the intensity profiles calculated by azimuthal integration along rings





(Fig. 13) and indicates that the "brown" original sample is less ordered than the "black" sample. This finding is also consistent with the Raman spectra and the real-space TEM images. Based on the comparison of the positions and profiles of the experimental peaks to the simulated ones it is concluded that all samples contain small graphitic-like atomic arrangements. The tendency of the maximum A to larger diffraction vectors by thermal treatment of the "brown" sample

refers to a slight reduction of the distance of graphitic layers by heating resembling that of the "black" one.

The quantitative evaluation of the peak width by the full width at half maximum (FWHM) is displayed for the different samples in Figure 14. In the case of the "brown" sample the FWHM of ring B decreases by 30% as a consequence of heating. This result can be interpreted as an increase of the coherently scattering domain size (Fultz et al., 2001) and an increased degree of structural order during heating. This interpretation holds also for the other peaks; e.g. the FWHM value

of peak A at 2.8 nm$^{-1}$ changes from 0.54 +/- 0.05 nm$^{-1}$ to 0.45 +/- 0.04 nm$^{-1}$ by heating up to 870°C (Fig. 13 and 14).

Based on these FWHM values and using the Scherrer equation (Fultz et al., 2001) the size of the coherently scattering domains (in the direction perpendicular to the (002) layers) changes from about 1.6 nm to 2 nm as a consequence of heating. On the other hand, the FWHM of the "black" sample does not change and neither does the internal structure of the material (Figure 13 and 14). "Black" original and heated samples were found to have coherently scattering domains with sizes of

about 2.2 nm from which we conclude a slight difference in the structural ordering between the "black" and the "brown" heated sample. The changes in the graphitic domain size of the brown sample during heating confirm the findings of the Raman measurements but the temperature dependence is still different. As shown in Fig. 14 the FWHM of the brown sample decreases already at temperatures below 870°C. This is in contrast to the Raman spectra, where a sudden change of the shape of the curves occurs at 870°C in He. The cause for this difference is not completely clear. A possible explanation might be a

reorientation and alignment of existing clusters and layers and therefore an increase of the size of coherently scattering domains at temperatures below 870°C. This would not necessarily change the Raman signal since the structure within the clusters can be kept unchanged and the existing clusters would only rotate, rearrange and align.

## 5.5 UV-VIS Spectroscopy

Figure 15 shows the wavelength dependence of light absorption by the "brown" and "black" original and heated samples.

The wavelength dependence of the "brown" sample changes during the heating process. The absorption spectra for the "black" heated and original samples, however, do not change significantly, particularly not in one distinct direction. The spectrum of the "brown" original sample has a strong wavelength dependence, but during the heating process this spectrum changes and becomes more and more similar to the spectra of the "black" samples.

The wavelength dependence of the absorption was fitted using the Angstrom power law:

$$\ln(I/I_0) = K * \lambda^{\alpha}, \qquad \text{(Eq. 2)}$$




where $I_0$ is the incident intensity, I the reflected Intensity, α the Angstrom exponent, λ the wavelength and K a constant. Figure 16 shows the absorption spectra for the "brown" original and heated samples with the spectrum for the "black" original sample for comparison. The Angstrom exponents of the respective curves are given in the figure.

The spectrum of the original "brown" sample starts with an Angstrom exponent of α=−1.79. During the heating process the
wavelength dependence decreases and reaches values of α≈-0.63 after the heating step of 870°C in He which is even lower than the Angstrom exponent for the "black" original sample (α=−0.92).

The low Angstrom exponent (−0.35) of the sample heated to 775°C in $O_2$ might suffer from measurement uncertainties. At this heating stage, most of the absorbing material has already been burnt off the filter and the K-M absorption signals are already very small.

This decrease of the Angstrom exponent for the "brown" sample indicates an increase of BC, which is consistent with the findings of the IS measurements. At the heating stage of 870°C in He and after this stage, the Angstrom exponent is even lower than for the black original sample.

This difference could have two explanations. The Angstrom exponent for the newly formed PC could be lower than for the original BC in the sample, or the Angstrom exponent of the original BC part in the "brown" sample could be lower. Both
would  result in the low Angstrom exponent of the heated sample.

**6 Conclusion**

From the various analyses a change in structure of the "brown" sample during thermal analysis can be clearly seen. For the "black" sample which consists of agglomerates of spherules with graphitic-like structure no such change occurs. These
findings are supported by all analysis techniques employed here (UV-VIS measurements, Raman spectroscopy, TEM and IS). The interpretation of the data of the "brown" sample, however, is quite complex and in some cases more than one interpretation might be possible. The Raman spectra show, that the structure of the "brown" sample changes significantly at 870°C in the He atmosphere. According to Ferrari and Robertson (2000) the relative increase of the D peak in the spectra indicates an increase of clustered aromatic rings in the material. Also the five curve fits of the spectra suggest that new small
graphitic layers could be formed during the 870°C heating step. The TEM measurements confirm the increase of ordering in the "brown" sample since the FWHM of the electron diffraction maxima e.g. at 4.9 nm$^{-1}$ decreases when the sample is heated. Consequently, an increase of domain size from 1.6 nm to 2 nm can be estimated for the original and heated "brown" sample, respectively. The FWHM, i.e. the level of ordering and the domain size, stays nearly constant over the subsequent cooling and re-heating process in the He+$O_2$ atmosphere. Although the FWHM value, the coherently scattering domain size and the layer distance at 870°C reach nearly the same values as those of the "black" sample, slight structural differences of
the "brown" and "black" sample are still present and can be e.g. related to the more perfect stacking of graphene-like layers in the "black" sample.



Likewise, the wavelength dependence of the UV-VIS spectra of the "brown" sample resembles the wavelength dependence of the "black" sample when the sample is heated to 870°C in He. The Angstrom exponent stays constant during the subsequent heating steps in He+$O_2$.

So all of the applied analysis techniques agree, that the structure of the "brown" sample resembles more or less the structure of the "black" sample, when the sample is heated to 870°C in He. All techniques also show that the structure does not change further during the subsequent heating steps.

For the last temperature step in the Helium phase (870°C) the results of the different measurement techniques applied in this study are consistent among themselves. However, the findings for samples heated at lower temperatures are somewhat contradictory: While the Raman spectra change suddenly at 870°C in He and do not change significantly during the preceding temperature steps (highest temperature 615°C in He), the decrease of the wavelength dependence and the darkening (i.e. the laser signal in the EC-OC analyzer) occur during the whole heating process. The change of the FWHM of the 4.9 nm$^{-1}$ of the electron diffraction maximum and the formation of BC also occur continuously.

It is not unambiguously clear, how the structural change proceeds during the first three heating steps. The optical measurement techniques suggest a continuous change, as the optical signals change continuously. The electron diffraction patterns also imply a continuous change. On the other hand, the Raman spectra clearly show, that Raman sensitive structures do not change significantly during the first three heating steps, which indicates that no new graphene layers (i.e. a clustering of aromates) are built. But the weakening of the wavelength dependence indicates a decrease of the optical band-gap in the material, which might be due to an increasing size of polyaromatic and graphene-like structures, i.e. larger conjugated π-molecular orbitals.

Oxygen and hydrogen can leave the sample at temperatures above 250°C in He (Petzold et al., 2013; Chow et al., 2004). The free bonds could be incorporated in the remaining molecule and form conjugated double bonds and increase aromaticity which would lead to the observed change in the UV-VIS spectra. Also oxygenated groups at the edges of the molecules have effects on the optical behaviour: a decrease of oxygen leads to a darkening of the substance i.e. weaker wavelength dependence. Both of these changes would explain the optical behaviour of the sample but would not significantly affect the Raman spectra of soot.

Even though it is not unambiguously clear, which differences in the structures lead to the conflicting results, our findings indicate that different physical-chemical processes occur at the lower temperatures in comparison to the 870°C (He) step. This is consistent with the findings of Yu and Yang (2002) who showed that the value of σ of PC in some cases is not constant during a single thermal analysis.

Overall we conclude, that the most significant and at the same time irreversible structural change of the pyrolyzed organic material in our samples happens at 870°C in He in the measurement protocol used here and that new graphene like layers are built. We can only speculate, however, on the causes of the continuous darkening of the sample during the lower temperature steps.




 **Acknowledgements**

This work was supported by the FWF, grant P26040. The Integrating Sphere Technique was developed within the grant H-85/92 Hochschuljubiläumstiftung der Stadt Wien. We thank Karin Wieland for the valuable advice for the Raman measurements.

**Author contribution**

Haller: conceptualization of the experimental setup, all measurements except TEM, evaluation and interpretation of data, preparation of MS

Rentenberger: TEM measurements, writing TEM section

10    Meyer: discussions and input for TEM measurements

Felgitsch: cooperation regarding UV-VIS measurements, writing part of the UV-VIS methods section

Grothe: discussions and input for the Raman and UV-VIS measurements, funding acquisition

Hitzenberger: conceptualization, supervision, discussions, extensive input to text





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





| Band | Shape | Position [$cm^{-1}$] | Vibration mode |
|---|---|---|---|
| $D1_{fit}$ | Lorentzian | ≈1360 | Disordered graphitic lattice ($A_{1g}$-symmetry), graphene layer edges |
| $D2_{fit}$ | Lorentzian | ≈1620 | Disordered graphitic lattice ($E_{2g}$-symmetry), surface graphene layers |
| $D3_{fit}$ | Gaussian | ≈1500 | Amorphous carbon |
| $D4_{fit}$ | Lorentzian | ≈1180 | Disordered graphitic lattice ($A_{1g}$-symmetry), polyenes, ionic impurities |
| $G_{fit}$ | Lorentzian | ≈1580 | Ideal graphitic lattice ($E_{2g}$-symmetry) |

**Table 1: Band positions and line shapes for the five fitting curves of soot Raman spectra (Sadezky et al. 2005)**

| CAST burning conditions | "Black" | "Brown" |
|---|---|---|
| Propane | 40 ml/min | 50 ml/min |
| Oxidation Air | 1.04 l/min | 0.9 l/min |
| Nitrogen | 0 | 0 |
| Dilution Air | 20 l/min | 10 l/min |
| Quench gas ($N_2$) | 7 l/min | 7 l/min |
| C/O | 0.275 | 0.397 |

**Table 2: CAST burning conditions used for the "black" and the "brown" sample. The dilution air refers to the CAST internal dilution.**




| Carrier gas | Temperature [°C] | Residence time [s] |
|---|---|---|
| He | 310 | 80 |
| He | 475 | 60 |
| He | 615 | 60 |
| He | 870 | 90 |
| He+$O_2$ | 550 | 45 |
| He+$O_2$ | 625 | 45 |
| He+$O_2$ | 700 | 45 |
| He+$O_2$ | 775 | 45 |
| He+$O_2$ | 850 | 45 |
| He+$O_2$ | 870 | 120 |

**Table 3: Temperature steps and residence times for the NIOSH870 protocol (Panteliadis et al., 2015; Birch and Cary, 1996)**

| | Thermal-optical measurement | Integrating Sphere Method |
|---|---|---|
| | EC/TC | BC/LAC |
| "Black" | 0.85 | 0.99 |
| "Brown" | 0.09 | 0.11 |

**Table 4: Results of thermal-optical and integrating sphere measurements.**



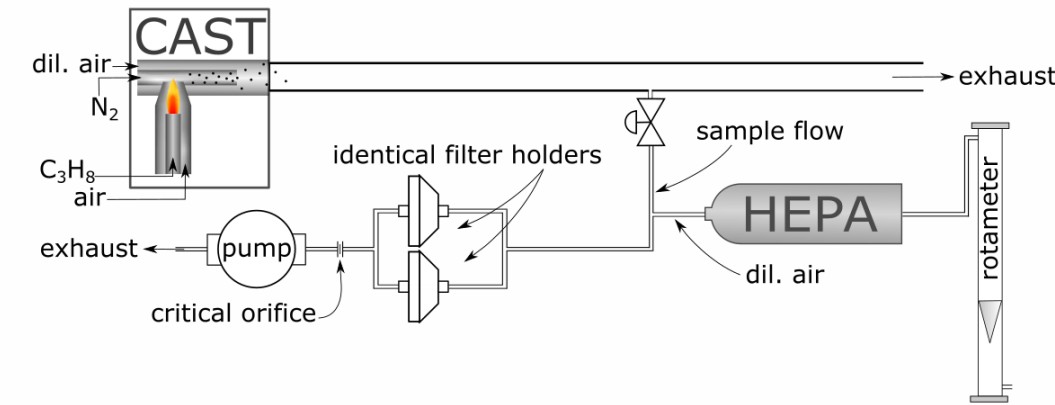

**Figure 1: Setup of the sampling. Soot produced by the CAST generator is first diluted within the generator. The sample flow is drawn from the exit line of the CAST generator and further diluted with filtered air (dilution flow regulated with a needle valve and measured with a rotameter). Aerosol is sampled in parallel on filters placed in two identical filter holders. Total flow in the sampling line is regulated by a critical orifice (12.5 l/min).**

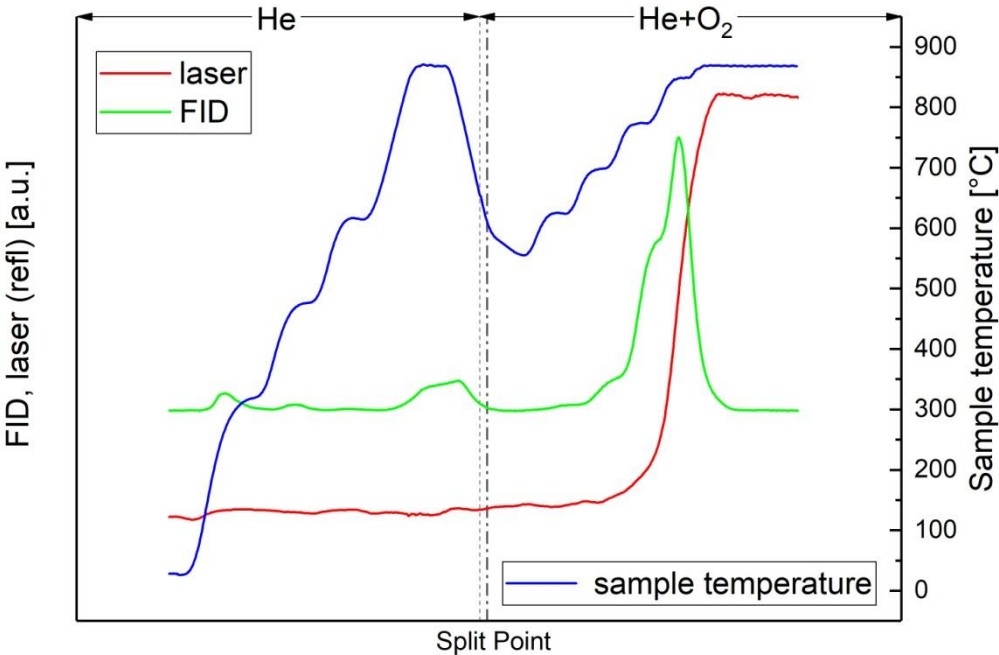

**Figure 2: Thermogram of the "black" sample. The blue line is the temperature measured at the position of the sample, the red line the laser reflectance signal and the green line the signal of the flame ionisation detector (FID), which is proportional to the amount of carbon leaving the filter. The split point was set at the point where the laser signal reached its initial value. The FID signal before the split point (broken line) is assigned to OC and after the split point to EC. The dotted line separates the inert and oxidizing phases.**





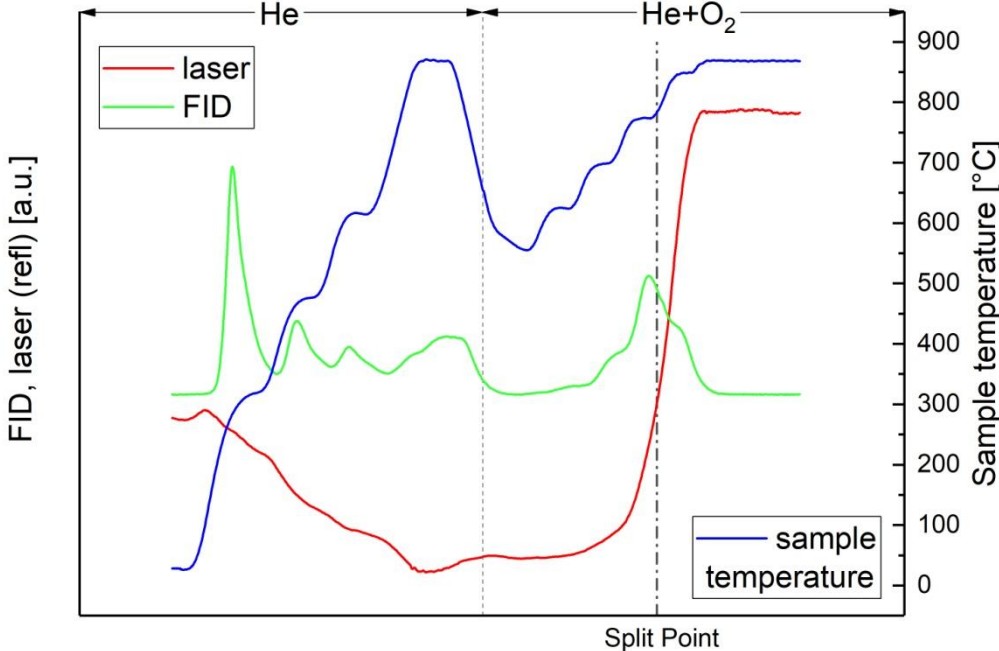

**Figure 3: Thermogram of the "brown" sample. The blue line is the temperature measured at the position of the sample, the red line the laser reflectance signal and the green line the signal of the flame ionisation detector (FID), which is proportional to the amount of carbon leaving the filter. The split point was set at the point where the laser signal reached its initial value. The FID signal before the split point (broken line) is assigned to OC and after the split point to EC. The dotted line separates the inert and oxidizing phases.**



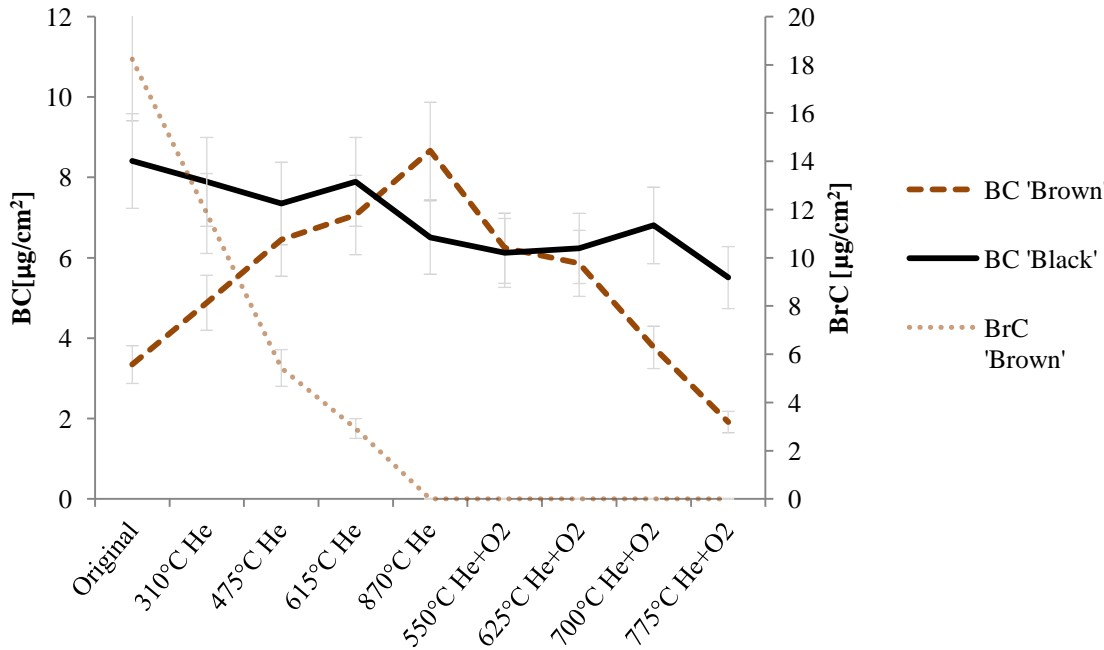

**Figure 4: Change of BC and BrC measured with the IS method during the heating according to the NOSH870 protocol measurement for the "black" and the "brown" sample. The error bars indicate deviations of ±14%. Laboratory intern test measurements gave variations in this range for filter samples measured in the integrating sphere.**




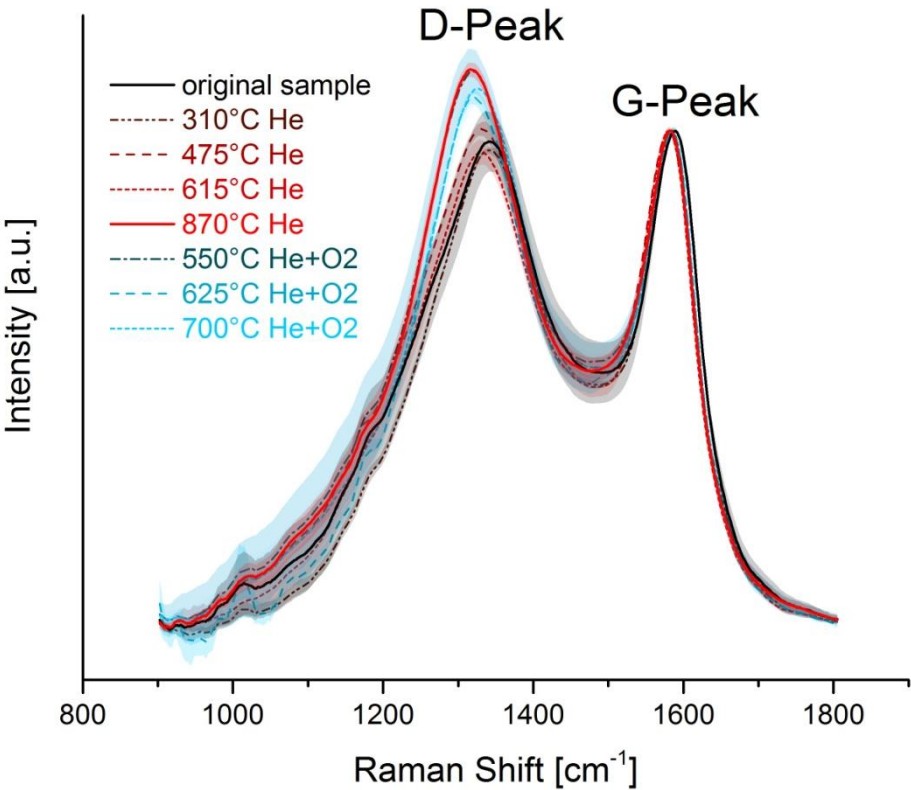

**Figure 5: Raman spectra of the original and the heated "brown" sample. The black and red solid lines represent the original sample and the sample heated to 870°C in He, where a significant structural change occurs. The mean spectra are smoothed and normalised to the maximum of the G-peak. The shaded areas represent the standard deviations of the mean spectra.**



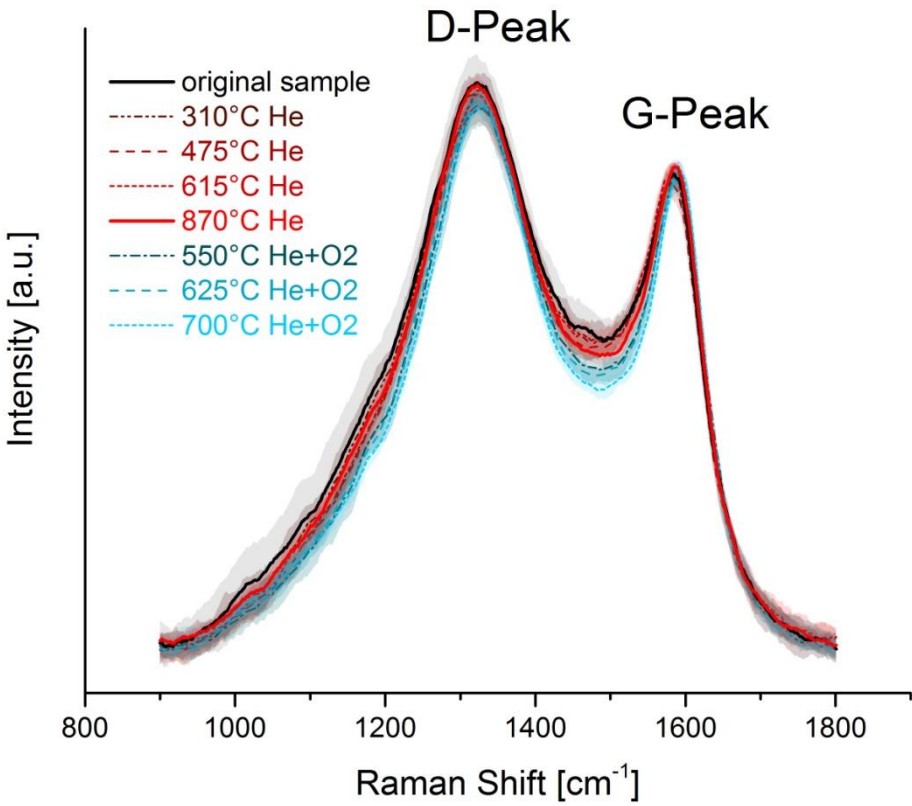

**Figure 6: Raman spectra of the original and the heated "black" sample. The black and red solid lines represent the original sample and the sample heated to 870°C in He. In contrast to the spectra of the "brown" sample, no significant change of the D-Peak height due to heating is visible here. The mean spectra are smoothed and normalised to the maximum of the G-peak. The shaded areas represent the standard deviations of the mean spectra.**





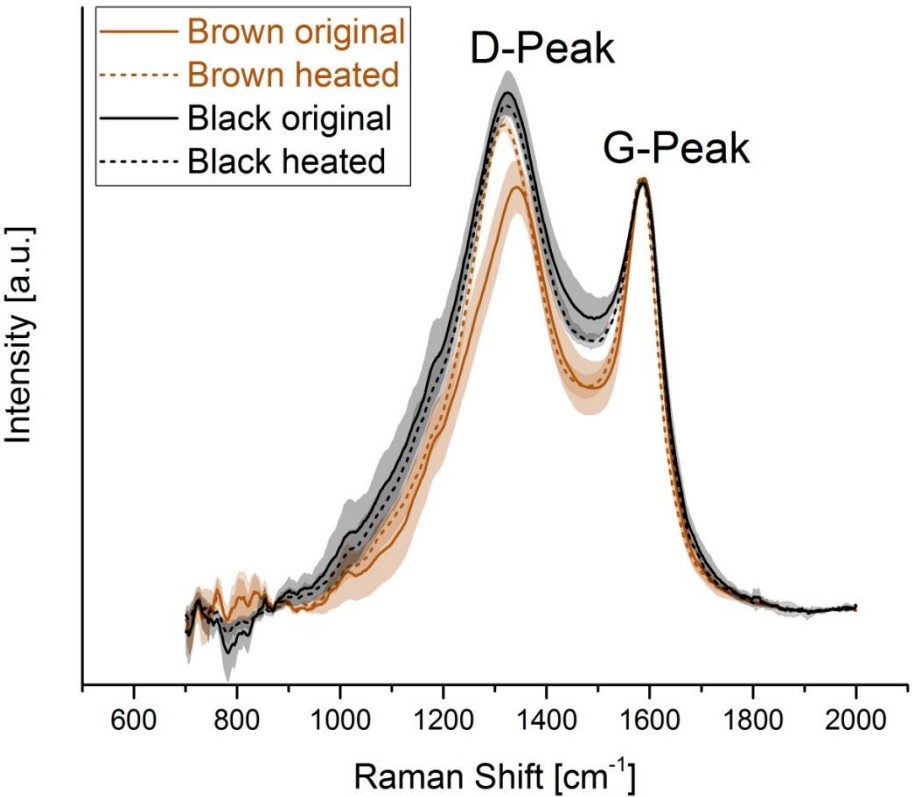

**Figure 7: Comparison of the Raman spectra of the original and heated (870°C He) samples for the "brown" and the "black" sample. The D-peak of the spectrum of the "brown" sample reaches the height of the D-peak of the black sample when the sample is heated to 870°C in He.**





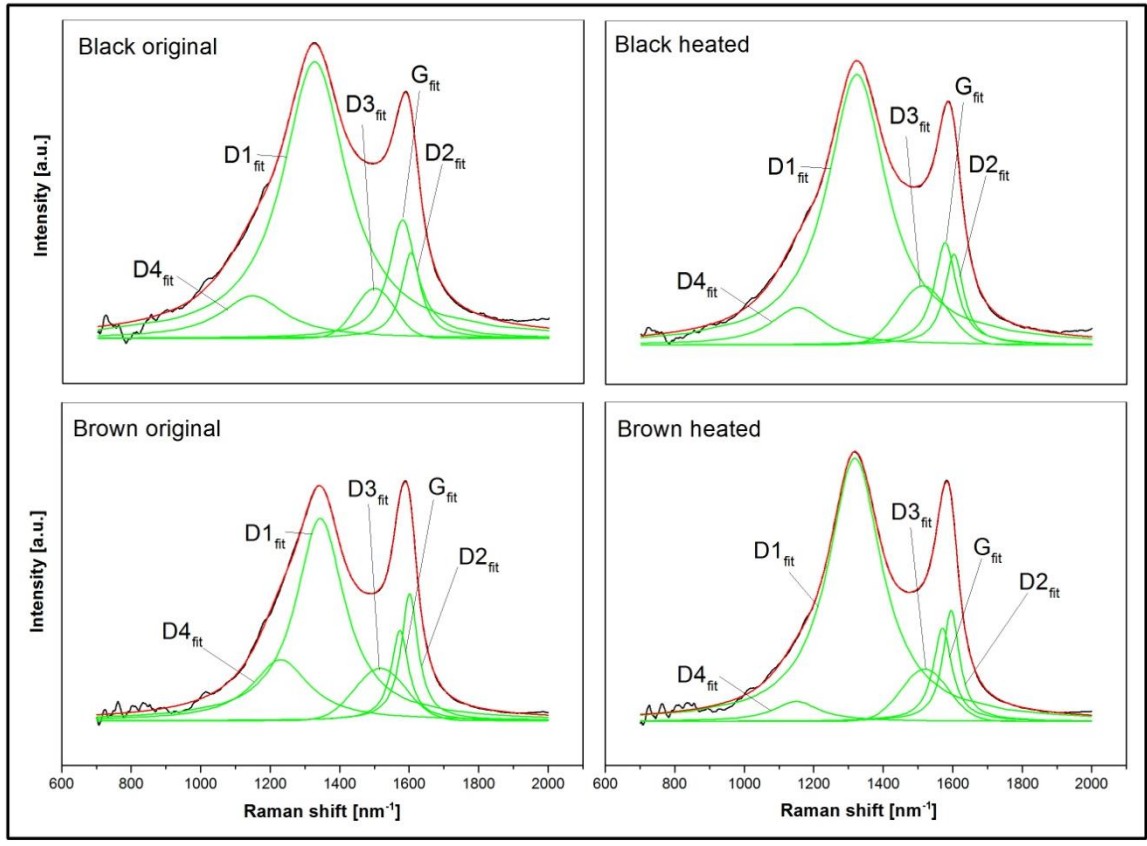

**Figure 8: Five curve fits of the original and heated (870°C He) spectra using the band positions and curve shapes of Sadezky et al. (2005). The black lines are the measured intensities, the green lines the five fitting curves (D1$_{fit}$-D4$_{fit}$ and G$_{fit}$) and the red lines the fitted spectra. The fitted curves of the "black" sample are similar before and after heating while the D1$_{fit}$ and D4$_{fit}$ of the "brown" sample change.**



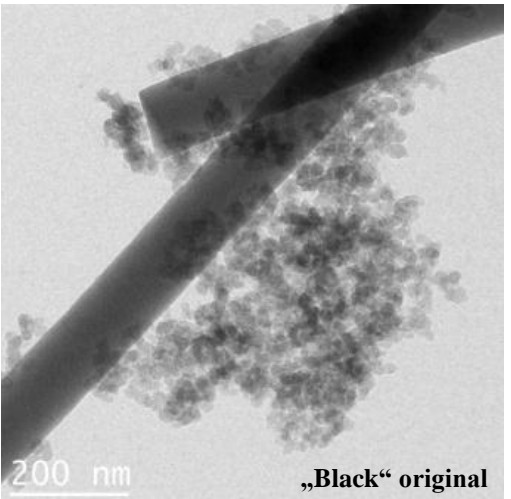
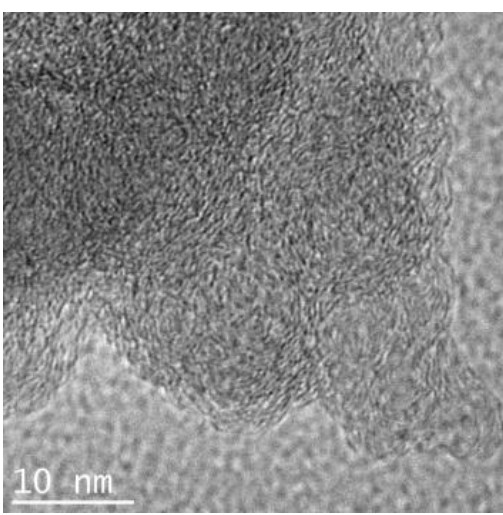

**Figure 9: TEM images of the "black" original sample with two different magnifications. The "black" sample consists of agglomerates of spherical primary particles. The higher magnification shows an onion like structure of the single spheres as reported by Sadezky et al. (2005) and Kim et al. (2015).**

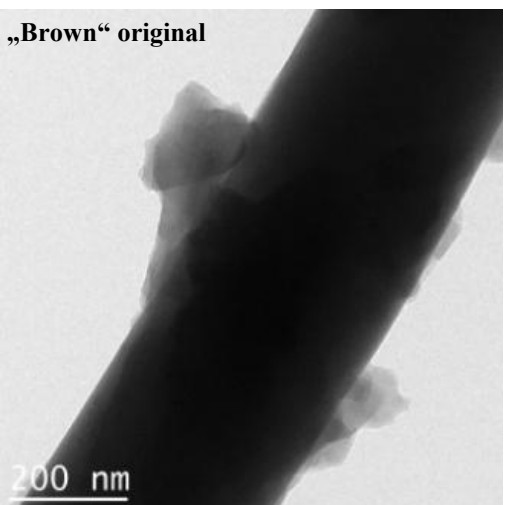
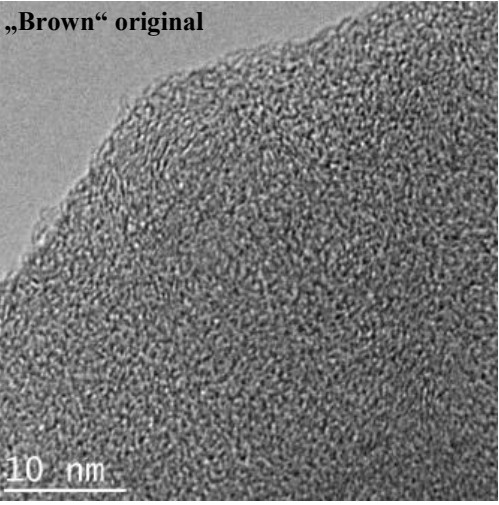

**Figure 10: TEM images of the "brown" original sample taken at two different magnifications. The "brown" sample consists of unstructured droplets. The disordered structure is visible in the right image.**



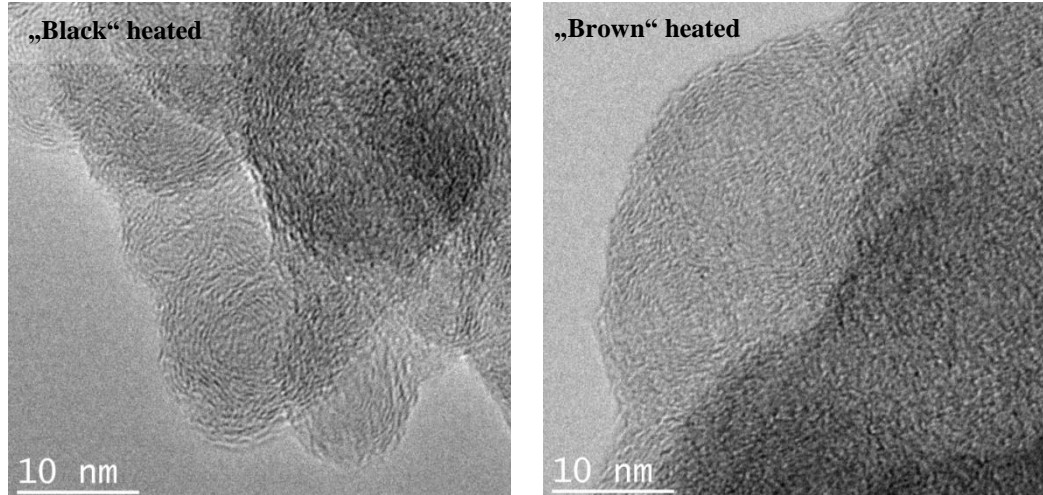

**Figure 11: TEM images of the "black" (left) and "brown" (right) heated (870°C He) samples taken at high magnifications. An increased ordering of the heated "brown" sample can be discerned, while the structure of the "black" heated sample does not change visibly in comparison to the original sample.**

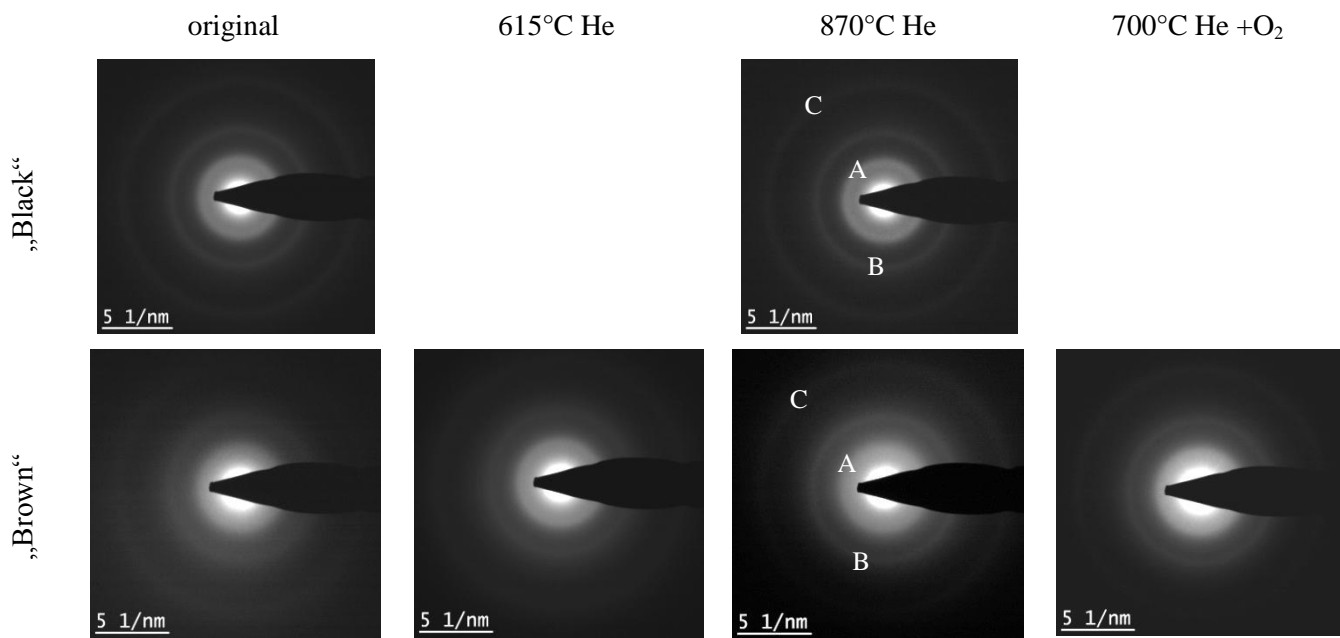

**Figure 12: Electron diffraction patterns of 'black' and 'brown' samples; original and heated. A, B and C indicate the ring maxima at 2.8, 4.9 and 8.4 nm$^{-1}$, respectively.**

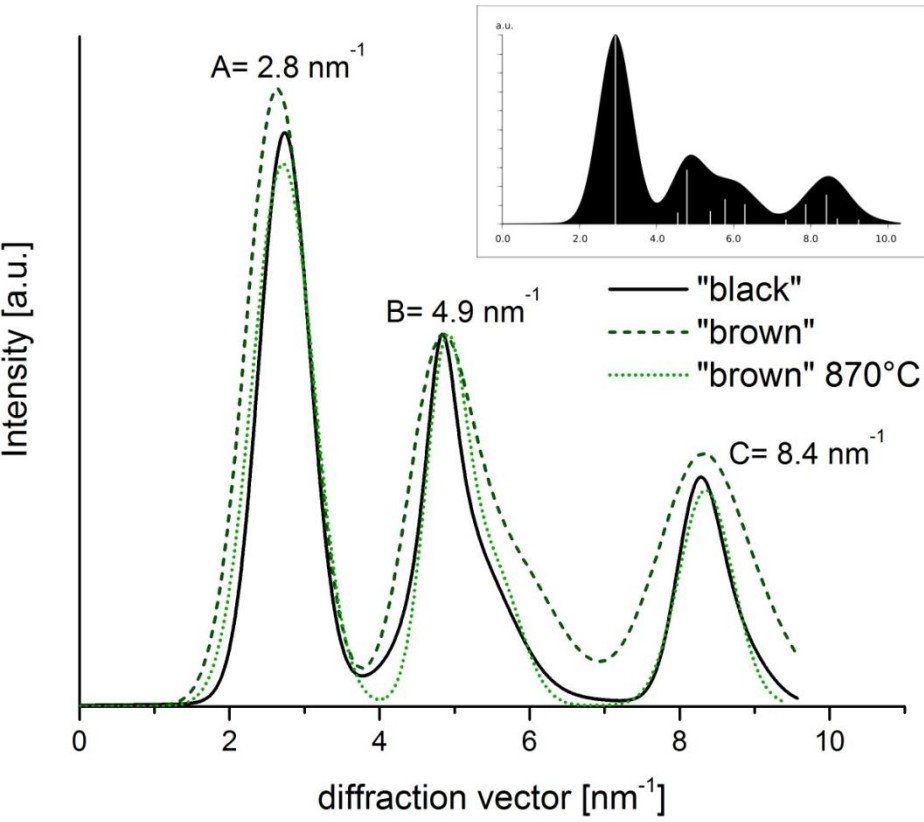

**Figure 13: Intensity profiles of diffraction patterns taken from the black and brown original sample and the brown heated (870°C He) sample. The intensities of all three samples were normalized to the peak intensity at 4.9 nm⁻¹. The image in the right corner shows the simulation for a graphitic material with small and randomly oriented graphitic clusters.**





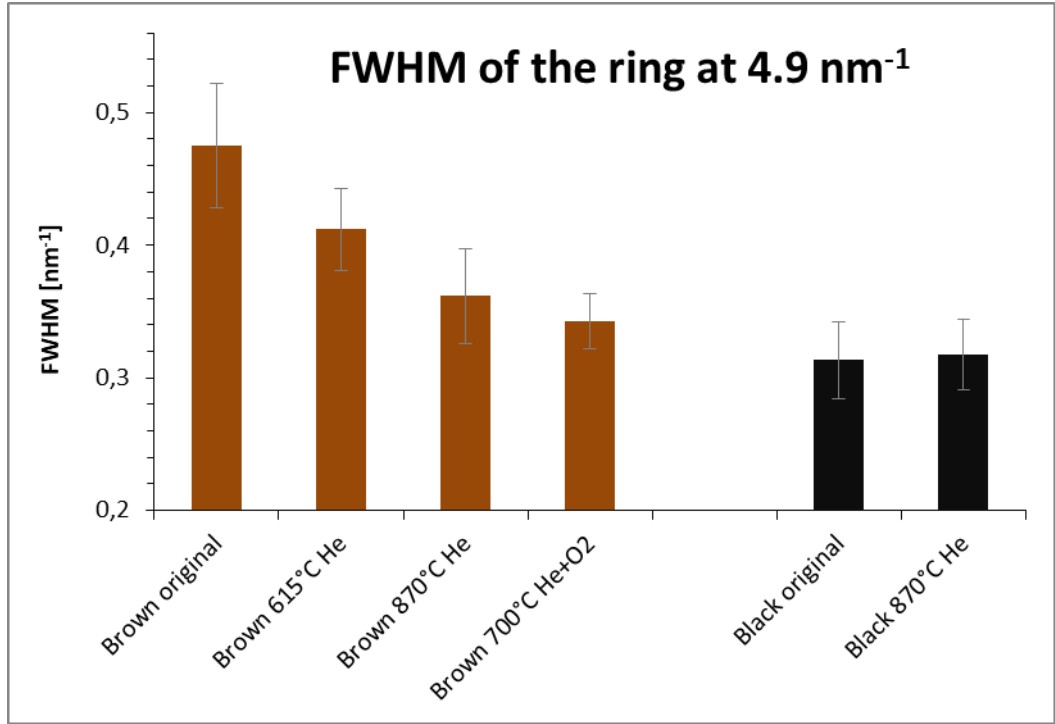

**Figure 14: FWHM of the 4.9 nm$^{-1}$ maximum of electron diffraction of the original and heated samples for selected temperatures. The FWHM of the "brown" sample decreases during the heating process and reaches values near to the FWHM of the black sample after heating at 870°C in He.**





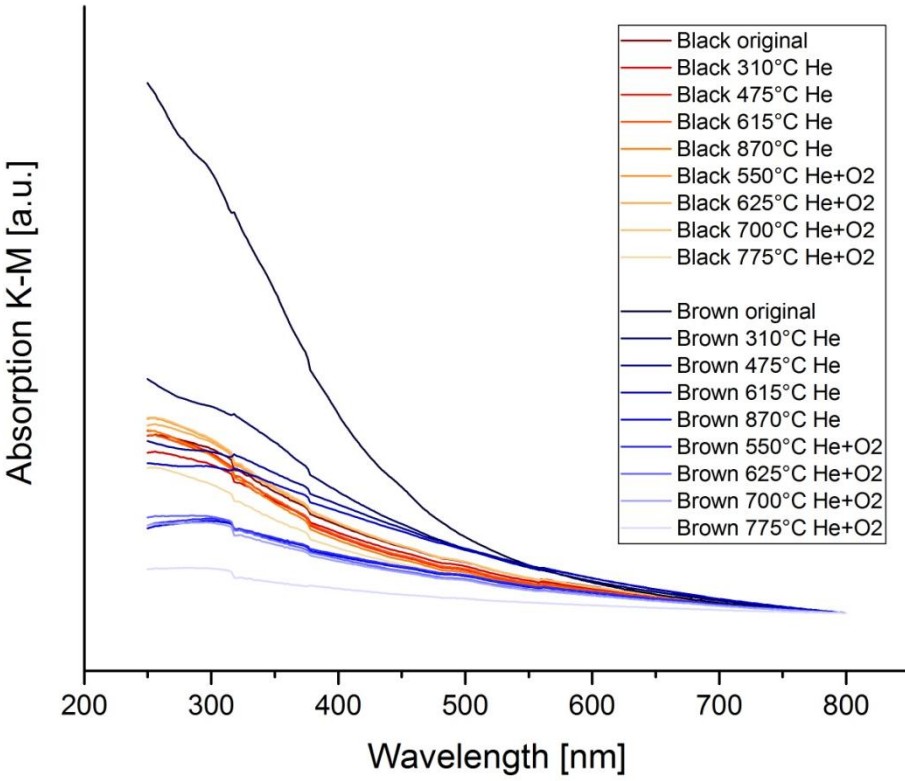

**Figure 15: UV-VIS spectra for the „black" and „brown" heated and original samples. The spectra are normalized at 800nm for better comparison. The wavelength dependence does not change significantly during the thermal-optical heating process for the "black" sample, while the wavelength dependence decreases continuously for the "brown" sample.**





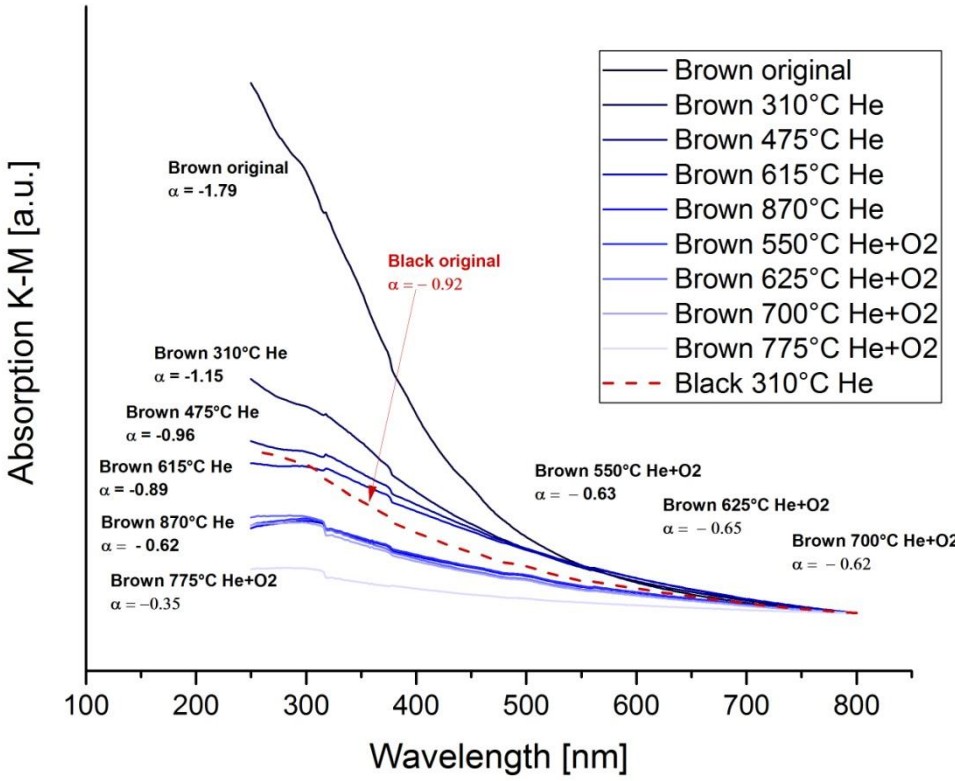

**Figure 16: UV-VIS spectra for the "brown" original and heated samples and corresponding Angstrom exponents. The dashed red line shows the spectrum of the "black" original sample for comparison.**