# Peer review of "Structural changes of CAST soot during a thermal-optical measurement protocol"

_Atmospheric Measurement Techniques, 2019_

## Referee Comment (RC1) · Darrel Baumgardner (Referee) · 7 Mar 2019

Before posting my limited remarks, I would like to congratulate the authors on a beautifully executed and documented study. The clarity of presentation, and the completeness of the analysis using multiple technique make this paper one that I plan to use to show students as a shining example of how to approach a problem, plan and execute the experiment, then document the results. In addition, the description of the different measurement techniques and the accompanying figures make these methodology sections ideal for educating the technically competent reader who does not have knowledge about these techniques.

One very minor suggestion concerning Fig. 4. Neither in the text or figure caption are

the two types of Brown carbon described that are shown by the brown lines. It took me a couple of minutes before I understood the difference.

A more substantive comment concerns the conclusions. I was expecting a summary discussion that would tie the results to the introductory problem statement, i.e. the difficulty in determining brown and black carbon concentrations when there are mixtures. Given the different measurement technique that were used to show how the brown carbon evolved as it was heated, if there potential for combining two or more of these technique to better improve the accuracy and decrease the uncertainty?

Or is this group already working on that concept for a followup paper?

---

## Referee Comment (RC2) · Anonymous Referee #2 · 12 Mar 2019

The paper reports on the study of the effect of thermal treatment of carbonaceous particle samples emitted from a miniCAST burner. The paper is well written and clear for the most part. I think the work is useful to the community and relevant, and I found the experiments and the data analysis and interpretation to be mostly appropriate; therefore, I recommend publication after minor revisions. A few comments follow.

General comments

The authors use "black carbon", "elemental carbon", and "soot" throughout the paper and the impression is that the term is used interchangeably. While that might be the case (I do not want to enter into the merit of that discussion here), it might be good to at least acknowledge that a related terminology problem is debated in the literature, see for example:

[Figure]

Buseck, P.R., K. Adachi, A. Gelencsér, É. Tompa, and M. Pósfai, ns-Soot: A Material-Based Term for Strongly Light-Absorbing Carbonaceous Particles. Aerosol Science and Technology, 2014. 48(7): p. 777-788.

Petzold, A., J.A. Ogren, M. Fiebig, P. Laj, S.M. Li, U. Baltensperger, T. Holzer-Popp, S. Kinne, G. Pappalardo, N. Sugimoto, C. Wehrli, A. Wiedensohler, and X.Y. Zhang, Recommendations for reporting "black carbon" measurements. Atmos. Chem. Phys., 2013. 13(16): p. 8365-8379.

The authors might also find interesting the following recent paper, in relation to the issue of charring:

Sedlacek, A.J., T.B. Onasch, L. Nichman, E.R. Lewis, P. Davidovits, A. Freedman, and L. Williams, Formation of refractory black carbon by SP2-induced charring of organic aerosol. Aerosol Science and Technology, 2018: p. 1-6.

Specific comments

Abstract: Please define IS, or better yet, considering the term integrating sphere is used only twice in the abstract, just spell it out.

Introduction, page 2, paragraph starting on line 12: Regarding the internal mixtures of carbonaceous and non-carbonaceous materials: several other published works might be of interest such as:

Adachi, K., S.H. Chung, and P.R. Buseck, Shapes of soot aerosol particles and implications for their effects on climate. Journal of Geophysical Research-Atmospheres, 2010. 115.

Adachi, K. and P.R. Buseck, Changes of ns-soot mixing states and shapes in an urban area during CalNex. Journal of Geophysical Research: Atmospheres, 2013. 118(9): p. 3723–3730.

China, S., C. Mazzoleni, K. Gorkowski, A.C. Aiken, and M.K. Dubey, Morphology and

mixing state of individual freshly emitted wildfire carbonaceous particles. Nature Communications, 2013. 4.

Cappa, C.D., T.B. Onasch, P. Massoli, D.R. Worsnop, T.S. Bates, E.S. Cross, P. Davidovits, J. Hakala, K.L. Hayden, B.T. Jobson, K.R. Kolesar, D.A. Lack, B.M. Lerner, S.-M. Li, D. Mellon, I. Nuaaman, J.S. Olfert, T. Petaja, P.K. Quinn, C. Song, R. Subramanian, E.J. Williams, and R.A. Zaveri, Radiative Absorption Enhancements Due to the Mixing State of Atmospheric Black Carbon. Science, 2012. 337(6098): p. 1078-1081.

And several others

Page 2, lines 34-35: papers of interest might also be the two papers mentioned above by Buseck et al. and by Petzold et al.

Page 5, lines 5 to 7: it might be useful to add a sentence or two on the fractal-like or lacy structure of the black carbon aggregates. Several works discuss this aspect in detail.

Page 6, line 12: Spell out "IS" as "integrating sphere" considering the acronym is defined only in a later section

Page 6, line 14: Briefly explain the role of the liquid. Why is it necessary?

Page 10, line 25-26: I find the sentence "However, the "brown" heated (870°C He) sample seems to have a more ordered internal structure in the form of layers than its original version as indicated by stronger noticeable local fringe-like contrast (Fig. 11)." Confusing, I thought the original brown sample had no ordered structure at all (as from line 22 on the same page). Similarly for the first line of page 11.

Line 5 and line 11 on page 12: what might be the physical or chemical reason for an Angstrom exponent lower than that of BC? The explanation in lines 13 to 15 is not very satisfying from a fundamental point of view.

―――――――――――――――――――

---

## Author Comment (AC1) · 7 May 2019

Darrel Baumgardner (Reviewer #1) darrel.baumgardner@gmail.com

Before posting my limited remarks, I would like to congratulate the authors on a beautifully executed and documented study. The clarity of presentation, and the completeness of the analysis using multiple technique make this paper one that I plan to use to show students as a shining example of how to approach a problem, plan and execute the experiment, then document the results. In addition, the description of the different measurement techniques and the accompanying figures make these methodology sections ideal for educating the technically competent reader who does not have

knowledge about these techniques.

Thank you for the appreciation of our work.

One very minor suggestion concerning Fig. 4. Neither in the text or figure caption are the two types of Brown carbon described that are shown by the brown lines. It took me a couple of minutes before I understood the difference.

We added a brief explanation in the figure caption describing the unclear abbreviations:

"BC "brown" and BrC "brown" refer to the amount of BC and BrC in the "brown" sample, BC "black" to the amount of BC of the "black" sample. BrC of the "black" sample is below detection limit for the original and the heated samples, respectively."

A more substantive comment concerns the conclusions. I was expecting a summary discussion that would tie the results to the introductory problem statement, i.e. the difficulty in determining brown and black carbon concentrations when there are mixtures. Given the different measurement technique that were used to show how the brown carbon evolved as it was heated, if there potential for combining two or more of these technique to better improve the accuracy and decrease the uncertainty? Or is this group already working on that concept for a followup paper?

Our main aim was to investigate the physical basis of the behavior of carbonaceous samples during the heating procedure of thermal-optical methods. We therefore used a soot generator which is widely used and produces samples that are rather well defined from a chemical point of view, i.e. that contain only carbonaceous material. This way we excluded the possible oxidizing effects of K+ and Na+ as well as sulfates which could occur in the He phase of the protocols. The paper shows the results of this investigation – the structural changes of the different samples are shown to our knowledge for the first time. These structural changes can, however, only be seen from TEM and Raman measurements, which are extremely time consuming and highly expensive both from the point of view of instrumentation and manpower.

Using these techniques routinely on the huge volume of filter samples produced in the measurement networks would be unfeasible.

Please also note the supplement to this comment:
https://www.atmos-meas-tech-discuss.net/amt-2019-10/amt-2019-10-AC1-supplement.pdf

―――――――――――――――――

---

## Author Comment (AC2) · 7 May 2019

Anonymous Reviewer #2

The paper reports on the study of the effect of thermal treatment of carbonaceous particle samples emitted from a miniCAST burner. The paper is well written and clear for the most part. I think the work is useful to the community and relevant, and I found the experiments and the data analysis and interpretation to be mostly appropriate; therefore, I recommend publication after minor revisions. A few comments follow.

General comments The authors use "black carbon", "elemental carbon", and "soot" throughout the paper and the impression is that the term is used interchangeably. While that might be the case (I do not want to enter into the merit of that discussion here), it

might be good to at least acknowledge that a related terminology problem is debated in the literature, see for example: Buseck, P.R., K. Adachi, A. Gelencsér, É. Tompa, and M. Pósfai, ns-Soot: A Material- Based Term for Strongly Light-Absorbing Carbonaceous Particles. Aerosol Science and Technology, 2014. 48(7): p. 777-788. Petzold, A., J.A. Ogren, M. Fiebig, P. Laj, S.M. Li, U. Baltensperger, T. Holzer-Popp,

S. Kinne, G. Pappalardo, N. Sugimoto, C. Wehrli, A. Wiedensohler, and X.Y. Zhang, Recommendations for reporting "black carbon" measurements. Atmos. Chem. Phys., 2013. 13(16): p. 8365-8379.

The nomenclature of BC EC and soot is a subject that has been discussed widely. In our paper, we do not use the terms interchangeably. EC is used for the analyte of the thermal-optical techniques and BC for the analyte of the IS technique. Soot is used only as an umbrella term for the output of the miniCAST burner, without any further specification. This nomenclature follows roughly the current consensus in the literature: The terms "EC" and "soot" are used as suggested by Petzold et al. 2013. The term "BC" is used for the "carbonaceous fraction with high light absorption in the whole visible wavelength range" (page 2, line 24), which is comparable to eBC of Petzold et al., 2013.

For a further clarification of the terminology we added a half sentence "although the terms are not the same since EC is defined by thermal and BC by optical properties of the material (Buseck et al., 2014; Petzold et al., 2013)." to line 34, page 2 and cancelled "respectively BC" in line 7, page 3. We changed line 18, page 4 to "The general term soot is usually used for particulate products of incomplete combustion [. . .]" and added in line 27 of the same page "[. . .] BrC, which is formed under oxygen poor and low temperature combustion conditions".

The authors might also find interesting the following recent paper, in relation to the issue of charring: Sedlacek, A.J., T.B. Onasch, L. Nichman, E.R. Lewis, P. Davidovits, A. Freedman, and L. Williams, Formation of refractory black carbon by SP2-induced

charring of organic aerosol. Aerosol Science and Technology, 2018: p. 1-6.

We had decided not to include this paper in our literature review, as it deals with charring in the SP2 instrument, while our study is entirely based on thermal-optical measurement techniques (in particular the NIOSH870 method). We now include a reference indicating that charring has also been observed in the SP2. We added a sentence in line 26, page 2: "Another distinction can be made between filter based techniques and techniques that analyse airborne particles in situ, such as photoacoustic spectrometry (Arnott et al„ 1999; Moosmüller et al., 1998) and the SP2 instrument (Stephens et al., 2003; Schwartz et al., 2006)." And in line 17, page 4: "Although charring of organic material has also been observed in SP2 measurements (Sedlacek et al., 2018), the present study is focussed on the processes occurring in filter samples."

Specific comments Abstract: Please define IS, or better yet, considering the term integrating sphere is used only twice in the abstract, just spell it out.

Changed as suggested: We inserted "(IS)" after "Integrating Sphere technique" in line 13, page 1.

Introduction, page 2, paragraph starting on line 12: Regarding the internal mixtures of carbonaceous and non-carbonaceous materials: several other published works might be of interest such as: Adachi, K., S.H. Chung, and P.R. Buseck, Shapes of soot aerosol particles and implications for their effects on climate. Journal of Geophysical Research-Atmospheres, 2010. 115. Adachi, K. and P.R. Buseck, Changes of ns-soot mixing states and shapes in an urban area during CalNex. Journal of Geophysical Research: Atmospheres, 2013. 118(9): p. 3723–3730. China, S., C. Mazzoleni, K. Gorkowski, A.C. Aiken, and M.K. Dubey, Morphology and mixing state of individual freshly emitted wildfire carbonaceous particles. Nature Communications, 2013. 4. Cappa, C.D., T.B. Onasch, P. Massoli, D.R. Worsnop, T.S. Bates, E.S. Cross, P. Davidovits, J. Hakala, K.L. Hayden, B.T. Jobson, K.R. Kolesar, D.A. Lack, B.M. Lerner, S.-M. Li, D. Mellon, I. Nuaaman, J.S. Olfert, T. Petaja, P.K. Quinn, C. Song, R. Subramanian,

E.J. Williams, and R.A. Zaveri, Radiative Absorption Enhancements Due to the Mixing State of Atmospheric Black Carbon. Science, 2012. 337(6098): p. 1078-1081. And several others

We added all suggested references in line 14, page 2.

Page 2, lines 34-35: papers of interest might also be the two papers mentioned above by Buseck et al. and by Petzold et al.

We added both references in line 37, page 2.

Page 5, lines 5 to 7: it might be useful to add a sentence or two on the fractal-like or lacy structure of the black carbon aggregates. Several works discuss this aspect in detail.

We inserted "fractal" in line 7, page 5 to point out also the fractal nature of agglomerated soot particles.

Page 6, line 12: Spell out "IS" as "integrating sphere" considering the acronym is defined only in a later section

Changed as suggested.

Page 6, line 14: Briefly explain the role of the liquid. Why is it necessary?

We inserted the following sentence in line 17:"Enhanced absorption caused by a possible coating effect is reduced by the liquid: Soluble coatings are removed from the particles and the effect of insoluble coatings is reduced because of the low relative refractive index of these coating compared to the liquid (Hitzenberger and Tohno, 2001)." Additionally we referred one more time to Hitzenberger and Tohno, 2001, where the role of the liquid is described more in-depth.

Page 10, line 25-26: I find the sentence "However, the "brown" heated (870_C He) sample seems to have a more ordered internal structure in the form of layers than its original version as indicated by stronger noticeable local fringe-like contrast (Fig. 11)."

Confusing, I thought the original brown sample had no ordered structure at all (as from line 22 on the same page). Similarly for the first line of page 11.

The sentence in lines 21-24, page 10 was changed to: "From the TEM images of the "brown" sample an ordered internal structure is not visible. The particles seem to consist of an oily substance [. . .]" and added: "However, analysis of the electron diffraction patterns indicates a small degree of ordering (see below). ", in lines 24-25.

Line 5 and line 11 on page 12: what might be the physical or chemical reason for an Angstrom exponent lower than that of BC? The explanation in lines 13 to 15 is not very satisfying from a fundamental point of view. We added the following paragraph in line 14-25, page 12: "Absorption Angstrom exponents depend on both the refractive index and on the size of the absorbing particles (Bohren and Huffman, 1983). Absorption by small particles (in the Rayleigh regime) has a stronger wavelength dependence than that by larger particles in the Mie scattering size range. The observed behavior of the Angstrom exponents could be caused by the different sizes and shapes of the particles of the "black" sample compared to particles in the "brown" sample. As we saw from the TEM images, the particles of the "black" sample consist of agglomerates of spherical primary particles with diameters of about 20 nm whereas the particles in the "brown" sample are larger than these primary particles. This large size difference is not substantially changed due to heating.

Please also note the supplement to this comment:
https://www.atmos-meas-tech-discuss.net/amt-2019-10/amt-2019-10-AC2-supplement.pdf
* * *